# QDC-2D: A Semi-Automatic Tool for 2D Analysis of Discontinuities for Rock Mass Characterization

**Lidia Loiotine** [1,2,*], **Charlotte Wolff** [1], **Emmanuel Wyser** [1], **Gioacchino Francesco Andriani** [2], **Marc-Henri Derron** [1], **Michel Jaboyedoff** [1] and **Mario Parise** [2]

1   Institute of Earth Sciences, University of Lausanne, 1015 Lausanne, Switzerland;
    charlotte.wolff@unil.ch (C.W.); emmanuel.wyser@unil.ch (E.W.); marc-henri.derron@unil.ch (M.-H.D.);
    michel.jaboyedoff@unil.ch (M.J.)
2   Department of Earth and Environmental Sciences, University of Bari Aldo Moro, 70125 Bari, Italy;
    gioacchinofrancesco.andriani@uniba.it (G.F.A.); mario.parise@uniba.it (M.P.)
*   Correspondence: lidia.loiotine@unil.ch or lidia.loiotine@uniba.it

**Abstract:** Quantitative characterization of discontinuities is fundamental to define the mechanical behavior of discontinuous rock masses. Several techniques for the semi-automatic and automatic extraction of discontinuities and their properties from raw or processed point clouds have been introduced in the literature to overcome the limits of conventional field surveys and improve data accuracy. However, most of these techniques do not allow characterizing flat or subvertical outcrops because planar surfaces are difficult to detect within point clouds in these circumstances, with the drawback of undersampling the data and providing inappropriate results. In this case, 2D analysis on the fracture traces are more appropriate. Nevertheless, to our knowledge, few methods to perform quantitative analyses on discontinuities from orthorectified photos are publicly available and do not provide a complete characterization. We implemented scanline and window sampling methods in a digital environment to characterize rock masses affected by discontinuities perpendicular to the bedding from trace maps, thus exploiting the potentiality of remote sensing techniques for subvertical and low-relief outcrops. The routine, named QDC-2D (Quantitative Discontinuity Characterization, 2D) was compiled in MATLAB by testing a synthetic dataset and a real case study, from which a high-resolution orthophoto was obtained by means of Structure from Motion technique. Starting from a trace map, the routine semi-automatically classifies the discontinuity sets and calculates their mean spacing, frequency, trace length, and persistence. The fracture network is characterized by means of trace length, intensity, and density estimators. The block volume and shape are also estimated by adding information on the third dimension. The results of the 2D analysis agree with the input used to produce the synthetic dataset and with the data collected in the field by means of conventional geostructural and geomechanical techniques, ensuring the procedure's reliability. The outcomes of the analysis were implemented in a Discrete Fracture Network model to evaluate their applicability for geomechanical modeling.

**Keywords:** rock mass characterization; discontinuity analysis; fracture network; point clouds; Digital Outcrop Models; identification of discontinuity sets

## 1. Introduction

In the last two decades, remote sensing techniques have become widespread and have attracted many researchers for their application on rock mass investigations. The main goal is to overcome the limits of the conventional geostructural and geomechanical surveys to characterize rock masses, related to logistics, limited access, vegetation, adverse weather conditions, and human and instrument errors [1–4]. In this perspective, terrestrial/aerial laser scanning and photogrammetry are widely accepted because of their capability to acquire high-resolution point clouds in a reasonable time and in safe conditions, with relatively low costs [5–7]. Great attention has been paid to discontinuities in 3D models,

given the key role they play in the mechanical behavior of rock masses and in affecting their stability [8–13]. With regards to engineering problems, the results of slope stability assessments by means of numerical modelling in discontinuous rock masses [14] depend on the reliability of the fracture characterization, which is fundamental to elaborate the rock mass conceptual model [15,16]. In the last decades, several methods were introduced in the literature to automatically or semi-automatically extract the discontinuity sets (DSs) affecting rock masses, as well as their properties, from raw or processed data derived from Light Detection and Ranging (LiDAR) [17–21], Structure from Motion techniques [5,22–26], or their combination [27,28]. As stated in [29], the extraction of the geometrical properties of discontinuities can be achieved by direct segmentation on point clouds [19,23,29–34] or by processing surfaces like Triangulated Irregular Networks (TINs) [3,20,35,36]. Each technique has its advantages and limits, which are highlighted in [37] in a detailed review of the different methods for the extraction of discontinuities and their properties from remote sensing products.

Nevertheless, flat or subvertical topographies in low-relief areas or in man-made excavations do not allow us to well detect the planar surfaces developed within rock masses from 3D models, since only the traces of the discontinuities are visible. As a result, information on mean direction, spacing, length, and persistence of the discontinuity sets could be misinterpreted and could lead to an improper characterization. In these circumstances, 2D analyses should be carried out on the traces of the discontinuities from orthophotos obtained utilizing LiDAR or SfM techniques.

Although methods to perform 2D analyses on discontinuities were published during the second half of the last century, they were not fully revised and implemented into a digital environment, to our knowledge. As a consequence, the data are usually collected in the field by means of time-consuming, locally dangerous, conventional geostructural and geomechanical surveys.

Some progress on this topic was achieved by means of commercial software/tools like ShapeMetrix 3D [38] or DiAna 2D [29] that require a license or are not available to the public. Concerning the open-source software and freeware, an estimation of the spacing and persistence of sets of discontinuities can be obtained using the Mattercliff software [39] on a photo by drawing straight lines (not polylines), while the up-to-date FracPaQ2D [40] is a useful tool to determine the strike and lengths of the traces, as well as the fracture intensity and density. In addition, the authors of [41] proposed a method to characterize Digital Outcrop Models (DOMs) by means of scan-line and scan-area analyses, which can be used as outcrop analogues to model reservoirs. This procedure identifies fracture parameters such as length, strike, intensity, density, and topology and enables localizing damage zones.

It is outstanding that, despite different techniques being introduced in the literature, a procedure for complete rock mass characterization has not yet been achieved [24,42–47].

To this aim, we developed a user-friendly, adjustable, and repeatable MATLAB routine to characterize the discontinuity sets and the fracture network of low-relief areas on DOMs by introducing the well-known formulas of the literature in a digital environment. Starting from an orthophoto, or a file with digitized traces, the routine calculates the trace orientations (strikes), lengths, spacing, and persistence; defines the fracture intensity and density; and estimates the block volume and shape, allowing us to characterize the investigated area in a few minutes. An innovative feature of the MATLAB routine consists of the semi-automatic identification of the discontinuity sets by means of two methods and the consequent classification of the traces, which can be chosen and validated by the user.

The routine was built on a synthetic dataset, which was specifically created, and validated on the orthophoto of a case study generated by applying the SfM on a dataset collected by Unmanned Aerial Vehicle (UAV) techniques.

Finally, the results obtained from the case study were used to generate a 3D geomechanical model of an adjacent and scarcely accessible sub-vertical rock cliff to assess the potentiality of the 2D analysis for engineering problems such as stability analyses. More

in detail, a stochastic Discrete Fracture Network (DFN) method was used to explicitly represent the discontinuity sets of the case study using the probability distributions of their orientation, spacing, length, and persistence obtained with the MATLAB routine. The DFN model creation shows how the proposed routine can be used to further investigate the geomechanical and hydrological behavior of rock masses through more realistic approaches. Further information on the DFN techniques, commercial codes, and numerical methods that integrate DFN techniques, which are out of the scope of this research, are reported in [48,49].

## 2. Materials and Methods

### 2.1. Study Site

The study area used to validate the routine is located in Polignano a Mare, on the Adriatic side of the Apulian coast (SE Italy) (Figure 1).

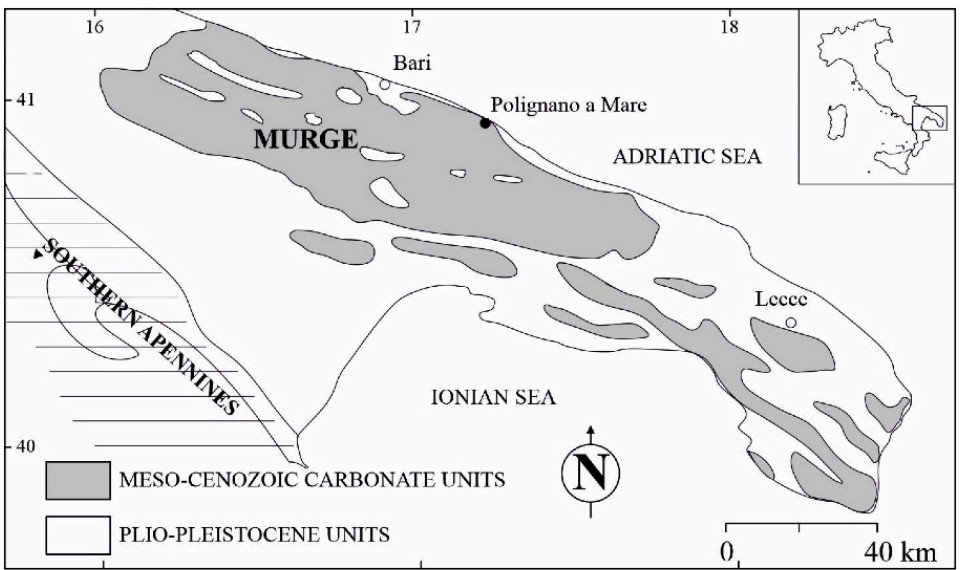

**Figure 1.** Geographic location of the study area.

Geomorphology of the area is characterized by a series of marine terraces subparallel to the coastline [50,51], gently dipping to NE and linked by small scarps. They are, at places, carved by water ways locally called lame [52]: slightly incised in the limestone bedrock and flat-bottomed, typically dry, valleys that constitute the main drainage network during exceptional rainfall events. The mentioned morpho-structures are the result of the superimposition of the tectonic uplift of the Apulian platform and the absolute sea-level changes, starting from the middle Pleistocene [53]. Platforms and cliffs form the coast up to 20 m high, linked by embayments constituted by coastal erosion deposits (pocket beaches).

From a geologic standpoint, the site belongs to the eastern part of the Murge plateau, an emerged block of the Apulian foreland characterized by a 3-km-thick Cretaceous succession related to a wide carbonate platform, overlain by upper Pliocene to Lower Pleistocene transgressive deposits of shallow marine waters [54,55]. The lithofacies outcropping in the study area are composed of whitish to greyish limestones and dolostones belonging to the Calcare di Bari Fm., which are discontinuously overlain by yellowish calcarenites belonging to the Calcarenite di Gravina Fm. While the latter has a massive structure, the former lithofacies is constituted by thin to medium bedded layers, crossed by a network of subvertical discontinuities and locally folded.

The fracture pattern, together with the marine and karst processes [56,57] strongly modelling this sector of Apulia, contributes to the geomorphologic evolution of the coastal area: Detachment niches along the subvertical walls of the cliffs and boulders at their base, occasionally visible below the sea level, indicate local failures of the rock mass, and further

potential instabilities cannot be excluded. These are among the most frequent geological hazards in coastal karst settings and are partly favored by the diffuse presence of karst conduits and caves, further weakening the carbonate rock mass [58–61].

Thus, a geostructural-geomechanical characterization of the site, with particular emphasis on the identification and characterization of the discontinuity sets and the estimation of the rock block shape, size, and volume, as well as the failure modes, is crucial for the management of prevention and mitigation measures at Polignano a Mare, one of the most important touristic sites of Apulia.

The rock mass considered for the 2D analysis is a 6400-m$^2$, low-relief platform developing from 10 m above the sea level to the current coastline, locally named Pietra Piatta. The choice of this area to perform the study was dictated by the lack of the Mediterranean vegetation and of anthropogenic elements as well, considering the disturbance produced by the residential center and by the vegetative area in nearby sectors. Due to erosional processes, the calcarenite facies crops out only in the external part of Pietra Piatta; therefore, a detailed view of the fracture traces in the Calcare di Bari Fm., which is generally covered by the Plio-Pleistocene deposits, is available at the site. In addition, 0.5- to 2-m steps located in the eastern side of the rock mass were found to be essential for the characterization of the bedding surfaces, which contribute to the formation of potentially unstable discrete blocks. Further, before implementing the study, field surveys were carried out to ensure that Pietra Piatta site was representative of the whole study area.

### 2.2. Generation of the Digital Outcrop Model (DOM)

A DOM of the study site was produced by means of Unmanned Aerial Vehicle (UAV) acquisition and Structure from Motion (SfM) processing techniques. The UAV survey at the study site was carried out in five steps:

(1) flight mission planning;
(2) positioning and coordinates' acquisition of Ground Control Points (GCPs);
(3) flight and image collection;
(4) Structure from Motion (SfM) processing and generation of the dense point cloud; and
(5) building of the orthomosaic.

To achieve the optimal coverage of the investigated area, an automatic flight mode, with front and side overlap of 75% and flight altitude of 18 m, was set in the planning phase. The nadir photogrammetric survey was performed on 12 December 2019, at early morning, in order to avoid sunlight reflections from the sea, using a quadcopter platform DJI Inspire 2 equipped with a 20.8-Megapixel (MP) resolution camera, an integrated satellite positioning system, and a remote flight controller. The system's specifications and the details of the survey are summarized in Table 1. In addition, three GCPs were manually positioned on the terrain before the flight such that they could be easily detected on the photos, and their coordinates were acquired by means of a Stonex SIII Differential Global Positioning System.

The SfM technique was carried out using Agisoft Metashape Professional [62] to process the images and obtain a 3D rock mass model. During the photo importation phase, the software automatically detected the camera calibration and location parameters (camera focal length, coordinates of the image principal point, and lens' distortion coefficients). The images were georeferenced in a WGS84/UTM 33 N metric coordinate system. The three GCPs were semi-automatically identified on the photos and their position was validated by the operator. Taking into consideration potential imprecisions in the acquisition of the GPS coordinates from the drone, these GCPs were used as a constraint to optimize the georeferencing of the model. The Root Mean Square Errors of the GCPs are reported in Table 2.

**Table 1.** Details of the UAV system, on-board camera, and photogrammetric survey.

| UAV System | |
|---|---|
| UAV device | DJI Inspire 2 |
| Maximum takeoff weight (g) | 4250 g |
| Maximum flight time (min) | 27 |
| Gimbal stabilization | 3-axis (pitch, roll, yaw) |
| On-board camera parameters and setting | |
| Camera model | Zenmuse X5S |
| Supported lens | DJI MFT 15mm 1.7 ASPH |
| Sensor | CMOS, 4/3″ |
| FOV | Effective Pixels: 20.8 MPx |
| Photo resolution (mm) | 72° |
| Survey details | |
| Flight mode | automatic |
| Ground Sampling distance (cm/pix) | 0.41 |
| Coverage area (km$^2$) | 0.836 |
| Flight altitude (m) | 18 |
| Number of photos | 248 |
| Front overlap (%) | 75 |
| Side overlap (%) | 75 |
| Frame shooting interval (s) | 1.5 |
| Ground resolution (mm/pix) | 4.71 |
| Number of tie-points | 311,321 |
| Number of projections | 2,290,325 |
| Reprojection error (pix) | 0.541 |
| GCPs XY error (m) | 0.010 |
| GCPs Z error (m) | 0.001 |
| Total GCPs error (m) | 0.010 |
| Orthomosaic pixel size (mm/px) | 4.71 |

**Table 2.** Root Mean Square Errors (RMSE) of the Ground Control Points used for the georeferencing optimization of the model.

| GCP ID | Number of Images | Horizontal Errors (cm) | | Vertical Errors (cm) | Total Error | |
|---|---|---|---|---|---|---|
| | | X | Y | Z | cm | pix |
| GCP1 | 32 | –0.78 | –0.03 | 0.02 | 0.78 | 3.065 |
| GCP2 | 22 | 1.19 | 0.74 | 0.21 | 1.42 | 0.744 |
| GCP3 | 18 | –0.40 | –0.71 | 0.14 | 0.83 | 3.374 |

Successively, the SfM algorithm recognized multiple key points in each picture and matched them in the overlapping photos. Then, 248/248 photos were aligned with the "high-accuracy" alignment option and optimized by means of sparse bundle adjustment algorithm [63], while the key point matches (tie points) were positioned in a 3D environment, thus obtaining a sparse point cloud. Later, a Multi-View Stereo (MVS) algorithm was applied to generate a "high-quality" 3D dense point cloud (98,375,478 points). The dense point cloud was cleaned of unwanted elements such as points belonging to the sea water moving on the borders of the model, and objects used to perform the photogrammetric surveys, by means of segmentation process. We chose to remove the disturbance in this phase because the creation of masks on the unwanted elements in the preliminary phase of the SfM processing would have required a large amount of time, considering the high number of photos.

The final step consisted of the generation of a mesh of the model, using the "high-quality" option available in Agisoft Metashape (9,785,754 faces) and in the extraction of a 2D orthomosaic from the mesh, with a 4.71-mm pixel size.

### 2.3. Conventional Characterization of Discontinuity Sets

Geostructural and geomechanical surveys were performed at Pietra Piatta site on 4 November 2020 to carry out a quantitative analysis of the discontinuities in the Calcare di Bari Fm. We chose to adopt window sampling techniques rather than scanline methods to avoid orientation biases, since discontinuities subparallel to the scanline are difficult to detect [8,64]. A preliminary visual inspection in the field helped to estimate the main discontinuity sets. Successively, two 100-m²-wide squares were created on the rock mass with a tape, to collect information on the discontinuities intersecting or contained in the survey window, following [65]. These areas, corresponding to two of the five sectors later analyzed with the MATLAB routine (sectors A and C in Figure 2), were selected for the presence of easily recognizable planar surfaces. Moreover, for each joint belonging to the analyzed discontinuity set, the strike, spacing, persistence, opening, and filling were measured by means of a Wilkie-type compass and a measuring tape. Roughness and wall strength were estimated using, respectively, a profilometer (Barton Comb) and a sclerometer (L-type Schmidt hammer). The strike data of the detected discontinuities were processed using the Dips software [66] by means of equal-angle stereographic projections (lower hemisphere) to identify the main DSs and calculate their statistical parameters. Successively, the geometrical data (i.e., spacing and trace length) collected in the field were processed on spreadsheets.

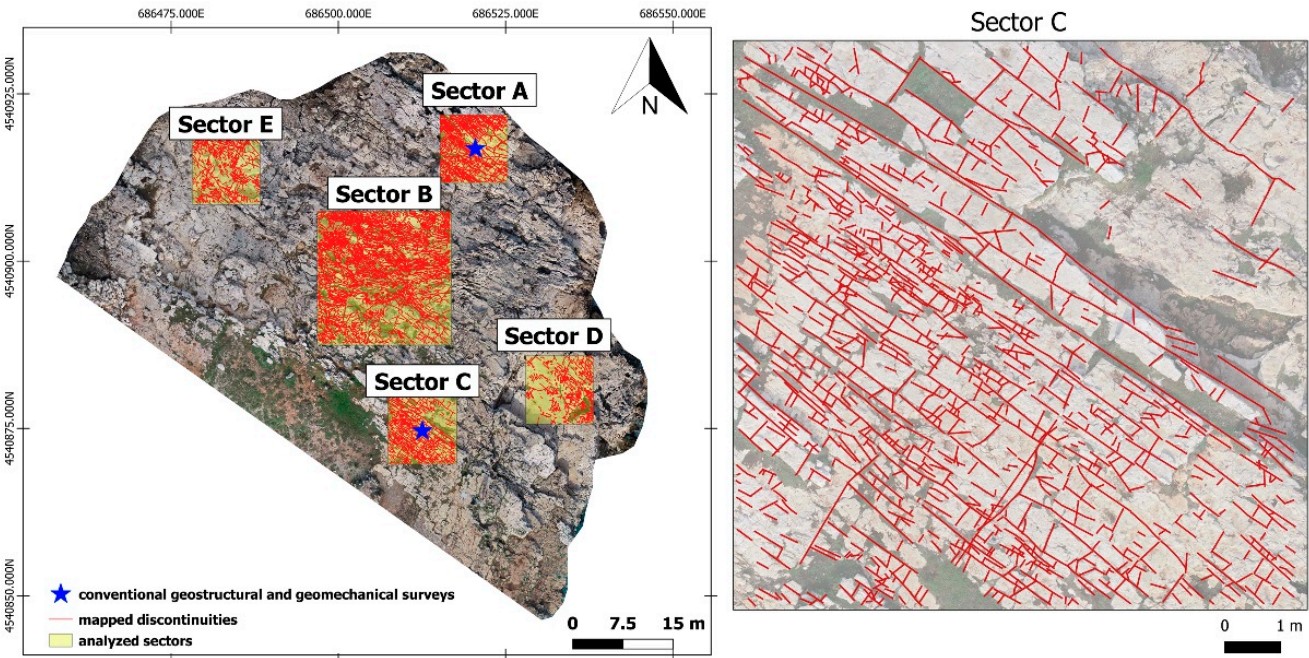

**Figure 2.** Sectors investigated by means of conventional geo-structural and geomechanical surveys and traces digitized in the orthophoto.

### 2.4. Manual Mapping of the Fracture Traces

The high-resolution orthophoto (1.25-GB-sized TIF image) produced by means of SfM technique was imported into a GIS environment by means of QGis open source software, in a WGS 84/UTM 33N metric coordinate system. After a meticulous visual inspection of the model, five sectors were considered to be representative of the rock mass; therefore, a vector layer made up of five square windows was created on the orthophoto. Successively, for each sector, the traces of the discontinuities were manually drawn as polylines keeping the same scale, which was established according to the resolution of the orthophoto, in order to reduce sampling biases in the different windows. The traces drawn on each polygon are reported in Table 3. The nodes of the polylines were extracted using the "extract vertices" geometry tool available in QGis, and their coordinates were obtained through the field

calculator in the attribute table. The "clip" vector algorithm (input: layer of the traces, overlay layer = square window) allowed us to extract the traces of each sector, as well as their nodes coordinates. Finally, the five layers were exported into CSV format and converted to text files with a spreadsheet.

**Table 3.** Number of traces drawn for each sector of the Pietra Piatta rock mass.

| Sector | Area (m$^2$) | Number of Traces |
|--------|------|------------------|
| A | 100 | 1472 |
| B | 400 | 5961 |
| C | 100 | 1018 |
| D | 100 | 818 |
| E | 100 | 1044 |

### 2.5. Methodological Approach for the Digital 2D Quantitative Analysis of Discontinuities

The code for the 2D quantitative analysis of discontinuities was written in MATLAB environment after a preliminary literature review of the discontinuity properties that must be defined for geostructural and geomechanical characterization of rock masses and of the methods to define them as well. According to [65], the quantitative description of the discontinuities requires the determination of their orientation, spacing, persistence, roughness, wall strength, aperture, filling, and seepage, as well as the number of discontinuity sets and the block size. In this study, we focused on the determination of the geometrical properties and on the number of discontinuity sets from orthophotos, derived from remote sensing techniques by implementing the methods presented in the literature in a digital environment. The parameters taken into account to perform the 2D analysis, the methods, and the source publications of the formulas are summarized in Table 4.

**Table 4.** Properties defined by the developed routine, applied methods, and relevant references.

| Property | Orientation | Normal Spacing | Frequency | Persistence | Trace Length Estimator | Intensity Estimator P$_{21}$ | Density Estimator P$_{20}$ | Block Volume | Block Shape |
|----------|-------------|----------------|-----------|-------------|------------------------|------------------------------|----------------------------|--------------|-------------|
| Method | Histogram Rose diagram | Scanline | | Window mapping | Circular window | | | From joint sets' spacing | |
| Reference | | [8] | | [67] | | [68,69] | | [70,71] | |

A synthetic dataset was created to gradually build the code. Three discontinuity sets (consisting of about 300 traces, in order to achieve statistical stability) were generated with strike and length following a normal distribution and a negative exponential law, respectively. The details of the synthetic dataset are illustrated in Figure 3. Successively, the 2D quantitative analysis was performed on the trace map (more than 1000 traces in sector C) of the case study and validated by comparing the results of the conventional geostructural and geomechanical surveys. To this aim, 50 discontinuities measured in the field in the same area were reported in Dips software [66] and classified into two main joint sets by means of stereographic projections (lower hemisphere). For each discontinuity set, the measurements of the spacings and trace lengths were elaborated in a spreadsheet to derive their mean values.

To test the potentiality of the 2D analysis for geological modelling, a Discrete Fracture Network model was generated on a 20-m-high cliff located at Lama Monachile site, about 90 m east from the study site. The top of the rock cliff is not visible because of the presence of the Calcarenite di Gravina Fm. and the overlying buildings. For this reason, the fracture pattern of the Calcare di Bari Fm. analyzed at Pietra Piatta site was used as an analogue model to generate the fracture sets by means of the FracMan software [72], after validation in the field of its representativeness for the cliff. The volume (grid) was generated from the

top and bottom surfaces of the mesh of the rock mass using a stochastic distribution of the bedding surfaces, which was determined from measurements on a vertical scanline on the point cloud. Successively, for each discontinuity set, the mean strike, standard deviation, minimum, maximum, mean spacing, and trace length derived from the MATLAB routine were used to generate the fractures in the DFN model.

| Name | Number of joints | Orientation (normal distribution) | | Trace length (negative exponential) |
|---|---|---|---|---|
| | | Mean ° | Standard deviation ° | Mean (m) |
| J1 | 100 | 75 | 10 | 2.0 |
| J2 | 80 | 100 | 7 | 1.0 |
| J3 | 125 | 120 | 5 | 0.8 |

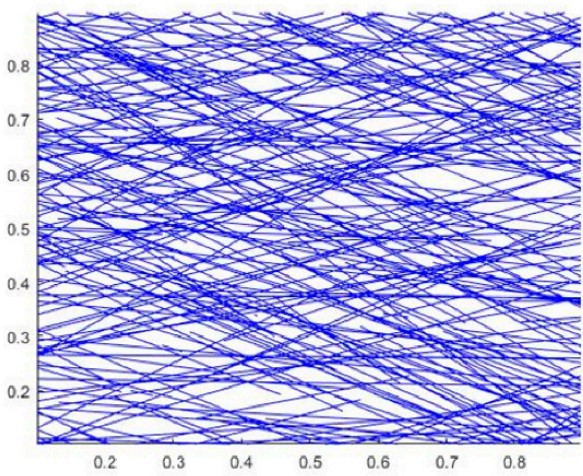

**Figure 3.** To the left, table illustrating the experimental dataset used to create the routine; to the right, graphical representation of the three joint sets.

### 2.6. Workflow of the MATLAB Routine

The MATLAB routine performs the 2D quantitative analysis on orthophotos by combining digital scanline and areal sampling methods. In detail, rectangular/square windows [8,67] are applied to identify the main discontinuity sets according to the orientation of the discontinuities, and to estimate their trace length and persistence as well. Circular windows [69,73] are performed to characterize the fracture network by identifying the mean trace length, intensity, and density estimators of the whole dataset. Areal sampling is preferred to scanline methods to avoid sampling inaccuracies such as orientation and length biases. In particular, orientation bias consists of the underestimation of the intensity of discontinuities, which are not perpendicular to a scanline [8,64,74], and length bias refers to the undersampling of short discontinuities with respect to the longer ones [75–77]. Since areal measures allow analyzing larger areas than scanlines [78], curtailment bias caused by the loss of information of discontinuities extending beyond the sampling windows [8] is also reduced. However, scanline techniques are indispensable to define 1D properties, such as normal spacing and fracture frequency of the discontinuity sets [8,79].

Aimed at creating a tool for a 2D quantitative analysis of discontinuities, available to the scientific community and easily reproducible on different case studies, the MATLAB routines are accessible through an editable template.

The 2D quantitative analysis is achieved in four steps (Figure 4):

1. graphical representation of the discontinuities;
2. semi-automatic classification of the discontinuity sets;
3. characterization of one discontinuity set; and
4. characterization of the fracture network.

The routine reads the first line of the template, in which the "STEP" command is defined. This allows performing the analyses described in Sections 2.6.1 and 2.6.2, by simply adding 1, 2, 3, or 4 next to "STEP". Precise instructions for the users are in the help and instruction files (Supplementary Material).

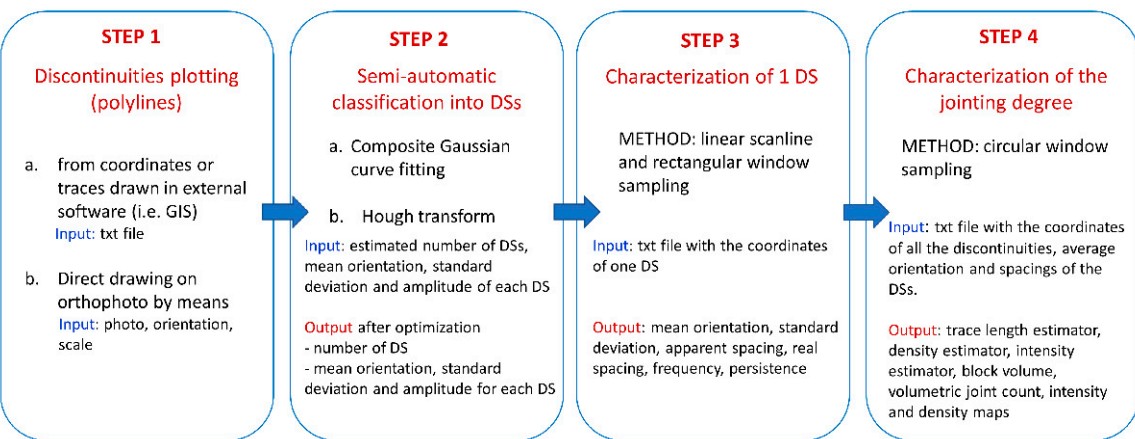

**Figure 4.** Step 1: workflow of the MATLAB routine developed for the 2D analysis of the discontinuities.

### 2.6.1. Characterization of the Discontinuity Sets
Orientation

This property is commonly expressed in terms of dip (maximum inclination of a discontinuity to the horizontal) and dip direction (direction of the horizontal trace of the line of dip, measured clockwise from north). However, since this study dealt with discontinuity traces mapped on an orthophoto, the orientation was given by their strike (trace of the intersection of an inclined plane with a horizontal reference plane).

Two methods are used to identify and classify the discontinuity sets according to the orientation of the traces.

- Method 1: fitting of the composite Gaussian curve

The orientations of the polylines are calculated from the strike of their segments and weighted according to their lengths by means of a square/rectangular window applied on the selected dataset. The number of discontinuity sets is identified by means of an innovative method, which semi-automatically plots the strike distribution of the traces on an appositely developed Graphic User Interface (GUI) (Figure 5). The MATLAB routine automatically extracts the distribution of each discontinuity set, with peaks corresponding to their mean strike. Three sliders allow adapting the strike distributions of each discontinuity set by interactively changing their mean, standard deviation, and amplitude. A line parallel to the x axis is used to filter out the "noise", derived from random discontinuities, which might be present in traces from real rock masses. After the manual adjustment of the curves, an optimization algorithm is run so that the sum of the curves and noise fits the raw orientation histogram curve. In detail, the best solution to fit the synthetic curve on the original curve is obtained by minimizing the differences between the curve of the real data and the experimental composite curve by means of the fminunc (Find minimum of unconstrained multivariable) algorithm, which is based on the BFGS Quasi-Newton method [80–83]:

$$\text{Raw \_histogram \_orientation} - \left(\sum \text{histo\_orientation}_i + \text{noise}\right) = 0 \tag{1}$$

where the two terms are the orientation histogram of the dataset and the sum of the experimental curves, respectively.

Successively, the MATLAB routine calculates the intersections between each normal curve, corresponding to the limits of the orientation of each joint set, to classify them. Therefore, for each DS, a file with the coordinates of the polylines' nodes is created.

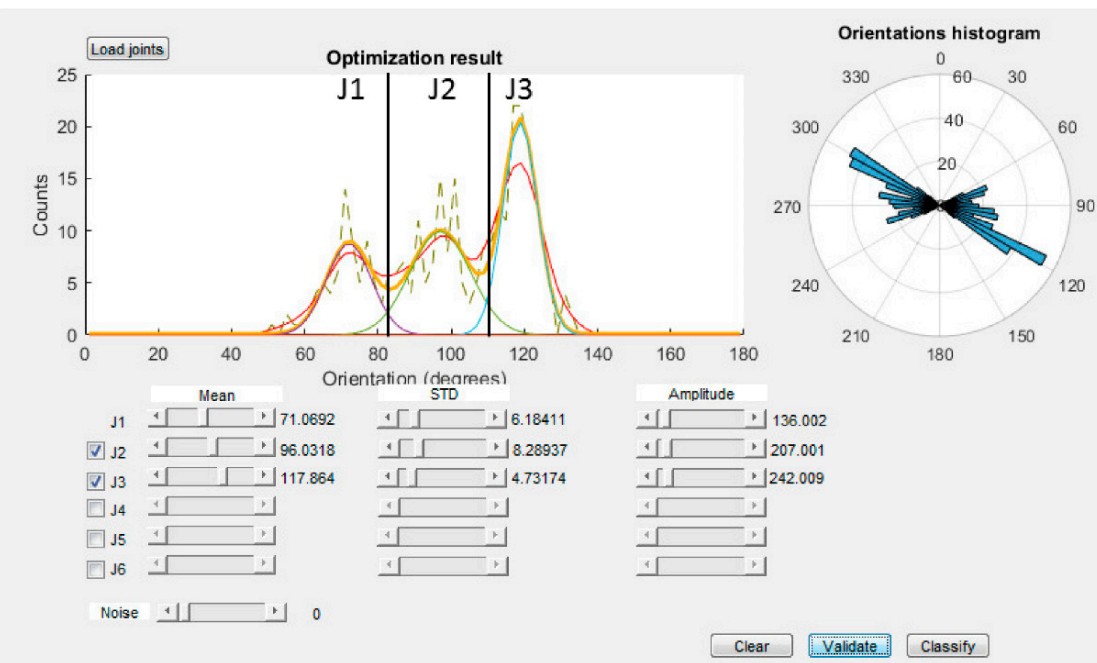

**Figure 5.** Step 1: Optimization process for the semi-automatic identification of the discontinuity sets forming the synthetic dataset by means of Gaussian composite curve fitting. The classification limits for each discontinuity set are represented by the black lines. The red and orange lines represent the smoothed curve and the estimation of the three joint sets, respectively; the purple, green, and blue curve represent J1, J2, and J3, respectively.

- Method 2: Hough transform

A second method to classify the discontinuity sets is based on the Hough transform, a technique introduced in [84] for machine analysis of bubble chamber photographs and later extended and adapted to image analysis and computer vision to detect shapes such as lines, circles, and ellipses in images [85–88]. Based on the Hough transform method, the MATLAB routine converts each joint in a new frame with the x and y axes corresponding to the orientation ($\theta$) and distance from the origin (r) of each polyline, respectively (Figure 6). In this new frame, each joint is defined by a point, and joints with similar orientation are aligned on the vertical axis, allowing us to identify the main discontinuity sets (Figure 7).

Spacing

The normal spacing of the discontinuity set (i.e., distance between two adjacent discontinuities belonging to the same set measured along a sampling line orthogonal to the mean direction of the set) can be calculated with two different methods according to their persistence. In a 2D analysis, the discontinuity persistence, which is the areal extent or size of a discontinuity within a plane [51], can be expressed as the limit length ratio along a given line on a joint plane [89]:

$$K = \lim_{L_S \to x} \sum_i \frac{l_{S_i}}{L_S} \tag{2}$$

where $L_s$ is the length of a straight line segment S and $l_{s_i}$ is the length of the i-th joint segment in S. In other words, persistent joints are represented by continuous traces, while non-persistent joints are formed by more segments separated by rock bridges.

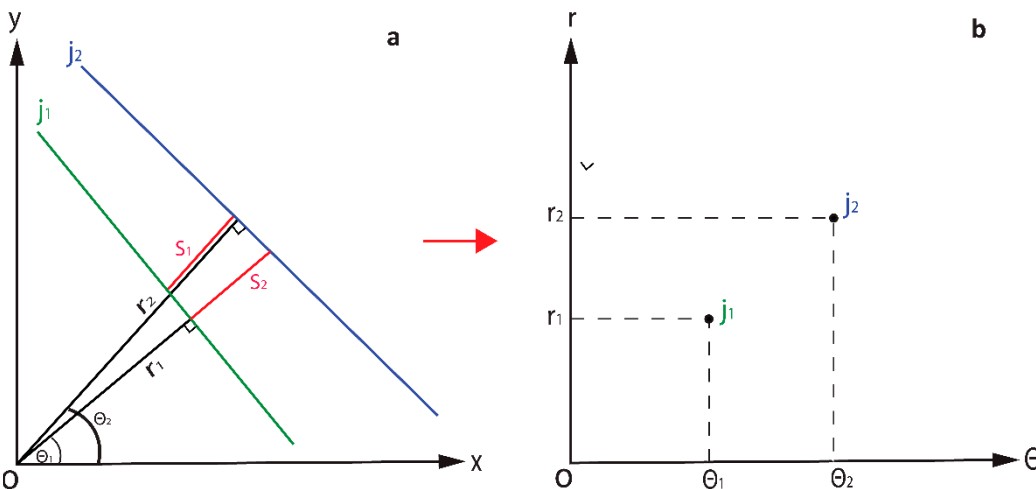

**Figure 6.** Conversion of discontinuity traces in the Hough frame. (**a**) representation of two discontinuity traces in the Cartesian reference frame, where r is the distance from the origin and θ is the angle orthogonal to the strike; (**b**) conversion into the Hough frame (θ,r): the lines are converted into points. The main discontinuity sets can be identified as points aligned on the r axis.

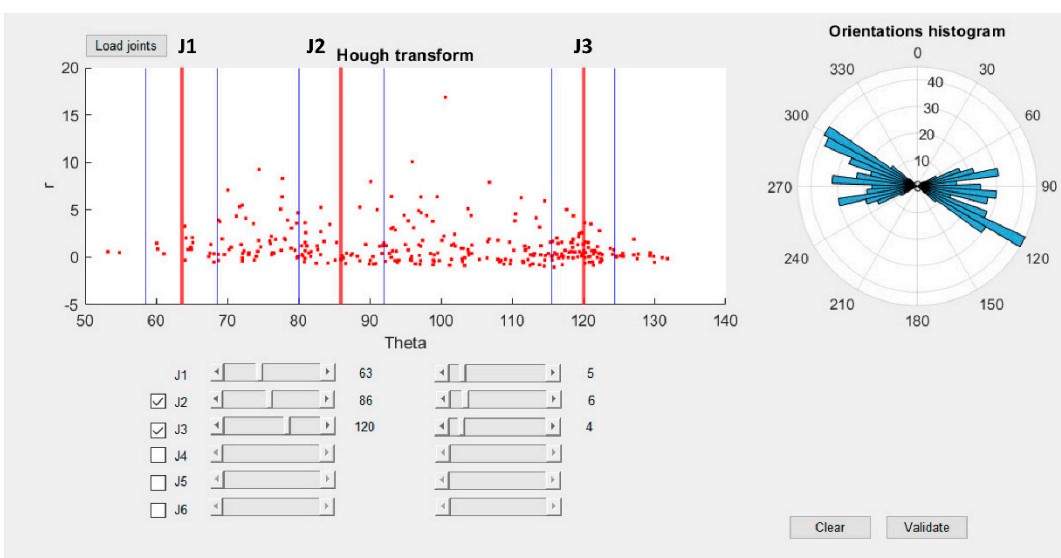

**Figure 7.** Application of the Hough transform for the classification into discontinuity sets.

- Spacing for non-persistent joints

After importing the file of one DS, the scanline method is applied to derive the normal spacing and frequency, with the assumption of non-persistent discontinuities. One or more scanlines can be plotted according to the user's setting. Specifically, one linear scanline can be automatically plotted in the middle of the frame orthogonal to the mean strike of the discontinuity set. To find the best scanline, the routine plots several randomly oriented scanlines and selects the one intersecting the highest number of joints. Alternatively, the user can pick the two endpoints or pick one endpoint and set the mean strike of the scanline directly on the trace map. In addition, a series of scanlines parallel to the reference one can be traced, with the possibility to choose the number of samples and the Δx–Δy interval. Considering the spatial variation of the strike of the traces, even if belonging to the same discontinuity set, Terzaghi's correction [64] is applied to each line intersecting the scanline to avoid orientation bias:

$$S = \frac{S_{app}}{\sin \theta} \quad (3)$$

where S, $S_{app}$, and θ are the corrected spacing, apparent spacing, and the minimum angle between the scanline and the mean strike of the discontinuity set. After the calculation of the apparent (not corrected) and normal spacings, the histogram and cumulative distributions of the spacings are plotted on the interface.

The normal spacing is then calculated as the mean distance of the intersections between the scanline and the traces (Figures 8 and 9). The mean fracture frequency is obtained as the inverse of the mean normal spacing.

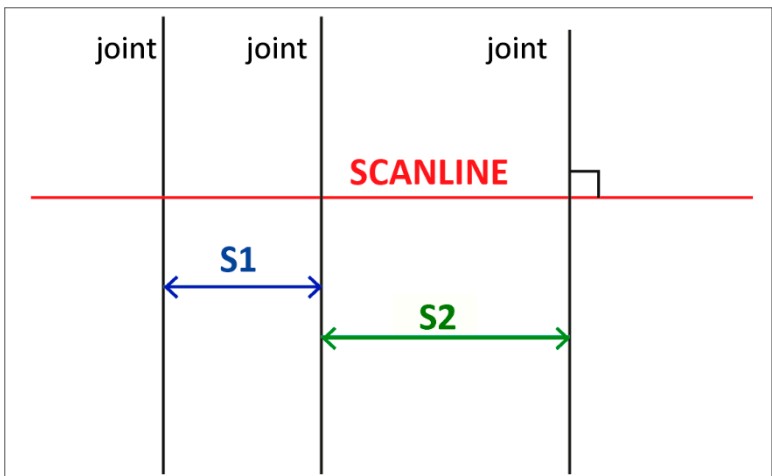

**Figure 8.** Normal set spacing calculated along a scanline (red) perpendicular to the mean direction of the joint set (black) (modified after [4]).

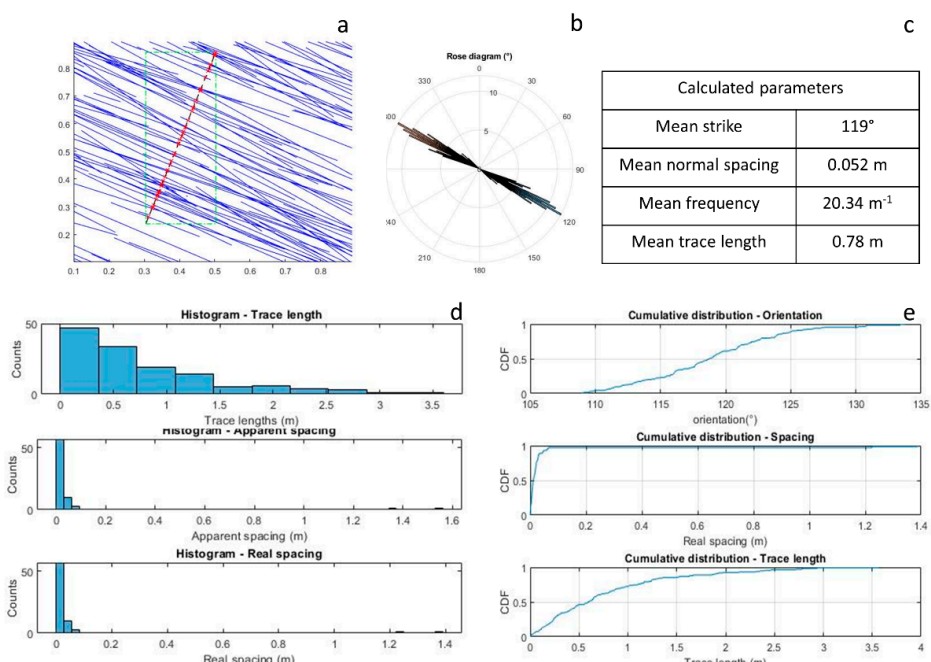

**Figure 9.** Step 3: linear scanline on JS3. (**a**) automatic generation of a linear scanline (black, dotted line) perpendicular to the mean strike of the discontinuity set; the discontinuities and their intersections with the scanline are, respectively, represented by the blue lines and the red crosses; (**b**) rose diagram of the JS; (**c**) table of the calculated parameters; (**d**) histograms of the calculated parameters; (**e**) cumulative distributions of the calculated parameters.

- Spacing for persistent joints

The Hough transform method is applied to calculate the spacing of persistent joints, assuming infinite lengths (Figure 6). For each two adjacent joints, the spacing is calculated as the mean difference along the r axis. In the Cartesian frame, the mean spacing is given by:

$$S = \frac{S_2 + S_1}{2} = \frac{[r_2 - r_1 \cos(\theta_2 - \theta_1)] + [r_2 \cos(\theta_2 - \theta_1) - r_1]}{2} \tag{4}$$

where $S_2$ and $S_1$ are the orthogonal distances between adjacent traces and $\theta_2$ and $\theta_1$ are the angles between the joints and the scanline.

Trace Length

The mean trace length of the discontinuity set is obtained by calculating the mean length of the polylines in the dataset. In addition, the histogram and cumulative frequency of the trace lengths are plotted on the interface.

Persistence

The method proposed in [67] is used to estimate the persistence of the discontinuity set by plotting a rectangular window on the trace map. The percentage of coverage area can be chosen by the user or drawn on the trace map. The discontinuities transecting the window $n_2$ (two intersections between the polyline and the window), contained in the window ($n_0$) (both endpoints are located in the window), and the total discontinuities ($n_{tot}$) are automatically counted, so that the mean persistence is calculated as:

$$\mu = \frac{w\,h}{w\cos\Phi + h\sin\Phi} \frac{n_{tot} + n_2 - n_0}{n_{tot} - n_2 + n_0} \tag{5}$$

where w and h are, respectively, the length and height of the window and $\Phi$ is the acute angle between the mean strike of the discontinuity set and the height of the window (Figure 10). The persistence is calculated in two steps:

(1)   over the entire dataset (Figure 11a);
(2)   by dividing the dataset with a grid and calculating the persistence for each element of the grid (Figure 11b,c).

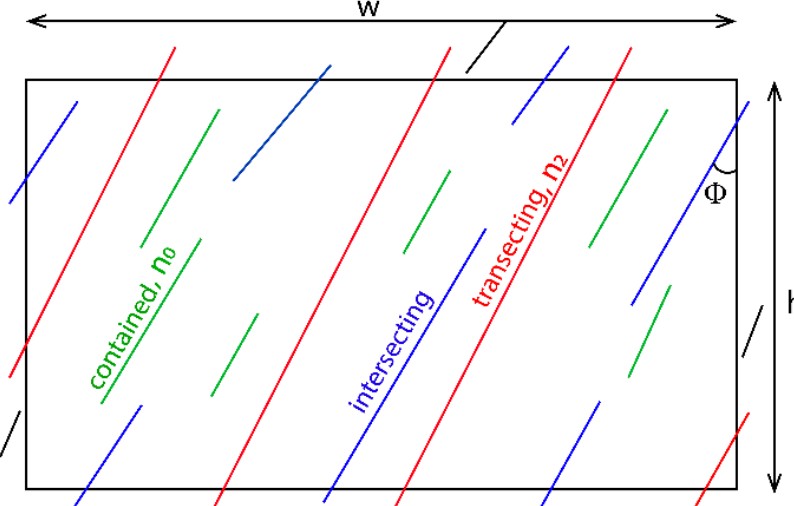

**Figure 10.** Method for persistence calculation (modified after [90]).

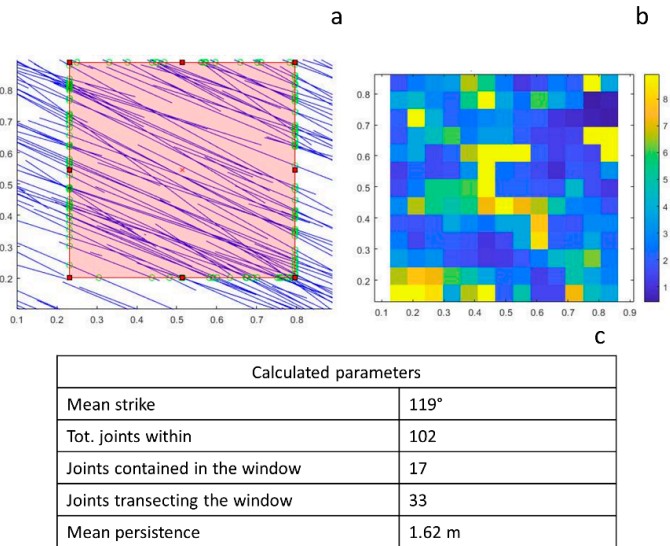

**Figure 11.** Step 3: persistence calculation for JS3. (**a**) persistence calculated over a rectangular window (colored in red). The intersection between the traces (blue) and the window are represented by the green circles; (**b**) persistence calculated for each element of the grid in which the window was subdivided; (**c**) results of the persistence calculation for the rectangular window.

The results of both methods are provided by the routine.

### 2.6.2. Characterization of the Fracture Network

Intensity, Density, and Trace Length Estimators

Information on the fracture network is collected through the method proposed in [91] by means of circular window sampling. A grid of circles is plotted on the polylines to compute the mean trace length $P_{11}$, the intensity $P_{21}$, and the density $P_{20}$ estimators:

$$\text{P11} \;=\; \frac{\pi r}{2}\,\frac{\bar{n}}{\bar{m}} \qquad \text{P21} \;=\; \frac{\bar{n}}{4r} \qquad \text{P20} \;=\; \frac{1}{2\pi}\frac{\Sigma m}{\Sigma r^2} \tag{6}$$

where $\bar{n}$ is the mean number of intersections between the polylines and the circles and $m$ and $\bar{m}$ are the total and mean number of endpoints of the polylines contained in the circles, respectively. Additionally, a persistence map can be plotted with the possibility of choosing the number of circles, with each pixel representing the persistence calculated in one circle.

Block Volume and Shape

Formulas illustrated in [70,71] are used to derive the block volume $V_B$, volumetric joint count $J_v$, and block shape factor β of the blocks delimited by discontinuities. As stated in [70,71], rock blocks are formed from the intersection of at least three discontinuity sets with different directions. Since rock volumes cannot be determined by means of 2D analyses from plan views, a third dimension is needed to apply this procedure. If only two joint sets are detected on the trace map, the third dimension can be represented by the bedding. If the rock mass is characterized by more than three discontinuity sets, a rough estimation of the volumes can be achieved by considering the prevailing ones.

The block volume can be estimated from the mean spacings of the discontinuity sets:

$$V_B = \frac{S1 S2 S3}{\sin \gamma_1 \sin \gamma_2 \sin \gamma_3} = \frac{S1 S2 S3}{\sin \gamma_3} \tag{7}$$

where S1, S2, S3 are the mean spacings of the three discontinuity sets delimiting the rock volume, and $\gamma_1, \gamma_2, \gamma_3$ are the angles between the discontinuity sets. Since the method is applied to joint sets perpendicular to the strata ($\gamma_1 = \gamma_2 = 90°$), only the angle between the two joint sets ($\gamma_3$) is taken into account.

For rock masses characterized by only one or two discontinuity sets, rock blocks can be formed when additional random joints cross the volume. In this case, the equivalent block volume can be calculated as:

$$V_B \approx 50\ S1^3 \text{ for only one joint set with mean spacing S1} \tag{8}$$

$$V_B \approx 5\ S1^2\ S2 \text{ for two joint sets with mean spacing S1 and S2} \tag{9}$$

The volumetric joint count is calculated as the number of joints per unit volume:

$$J_v = \Sigma \frac{1}{S_i} \tag{10}$$

where $S_i$ is the mean spacing of the i-th discontinuity set. The third dimension is needed to estimate $J_v$.

The parameters $\alpha_2$ (medium spacing/smallest spacing) and $\alpha_3$ (largest spacing/smallest spacing) are automatically calculated to determine the block shape factor β and to plot it on the chart shown in [70,71] in order to classify the block shape:

$$\beta = \frac{(\alpha_2 + \alpha_2\alpha_3 + \alpha_3)^3}{\alpha_2\alpha_3{}^2} \tag{11}$$

## 3. Results

### 3.1. Results of the Synthetic Dataset

The synthetic dataset was classified into three discontinuity sets by means of the Hough Transform and composite Gaussian curve fitting (Tables 5 and 6).

**Table 5.** Main parameters for the generation of the synthetic dataset and classification of the joint sets by means of Gaussian composite curve fitting.

| | Synthetic Dataset | | | Results of the Classification-Gaussian Fitting | | | |
|---|---|---|---|---|---|---|---|
| Name | Mean Strike | St. Deviation | N. of Joints | Name | Mean Strike | St. Deviation | Amplitude |
| J1 | 75° | 10° | 100 | J1 | 71° | 6° | 136 |
| J2 | 100° | 7° | 80 | J2 | 96° | 8° | 207 |
| J3 | 120° | 5° | 125 | J3 | 118° | 5° | 242 |

**Table 6.** Results of the classification of the joint sets by means of Hough transform method.

| | Results of the Classification-Hough Transform | | |
|---|---|---|---|
| Name | Mean Strike | Minimum Strike | Maximum Strike |
| J1 | 63° | 53° | 73° |
| J2 | 86° | 44° | 98° |
| J3 | 120° | 112° | 128° |

Figures 9 and 11 illustrate, respectively, the results of the scanline and rectangular window methods on the discontinuity set JS3 to calculate the mean orientation, normal spacing, trace length, and persistence.

The mean intensity, density, and trace length estimators of the synthetic dataset, calculated by means of circular windows, are 177.15 m$^{-1}$, 25.38 m$^{-2}$, and 1.14 m, respectively. A graphical representation of the results is given by means of intensity and density maps (Figure 12).

The block volume, volumetric joint count, and shape factor were calculated as 0.01 m$^3$, 33.81 m$^{-1}$, and 40.58, according to the mean strike and normal set spacings. Moderately flat blocks were identified on the diagram in Figure 13.

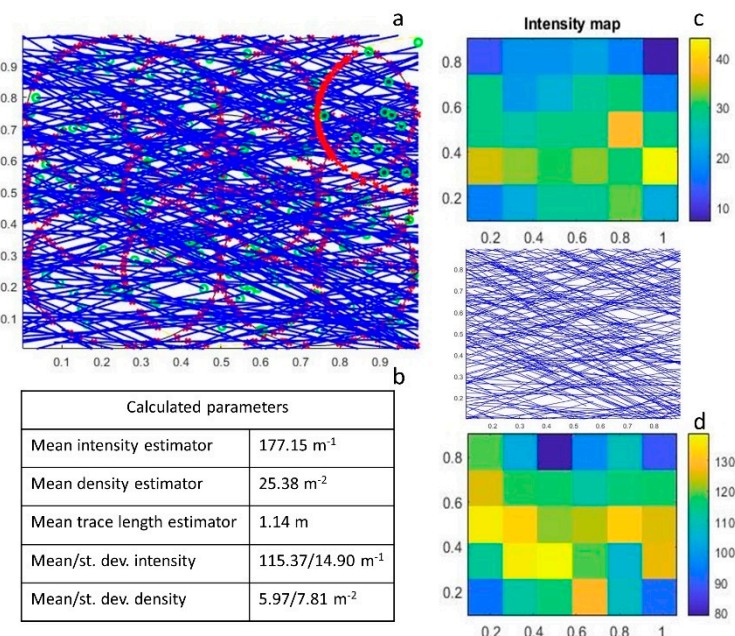

**Figure 12.** Step 4: generation of circular windows for the calculation of mean intensity, density, and trace length estimators of the whole dataset. (**a**) graphical representation of the circular windows, where the extremities of the traces transecting the circles are represented in green, and the traces intersecting the window (one intersection) are marked by the red circles. (**b**) Table of the calculated parameters; (**c**) intensity and (**d**) density maps of the dataset.

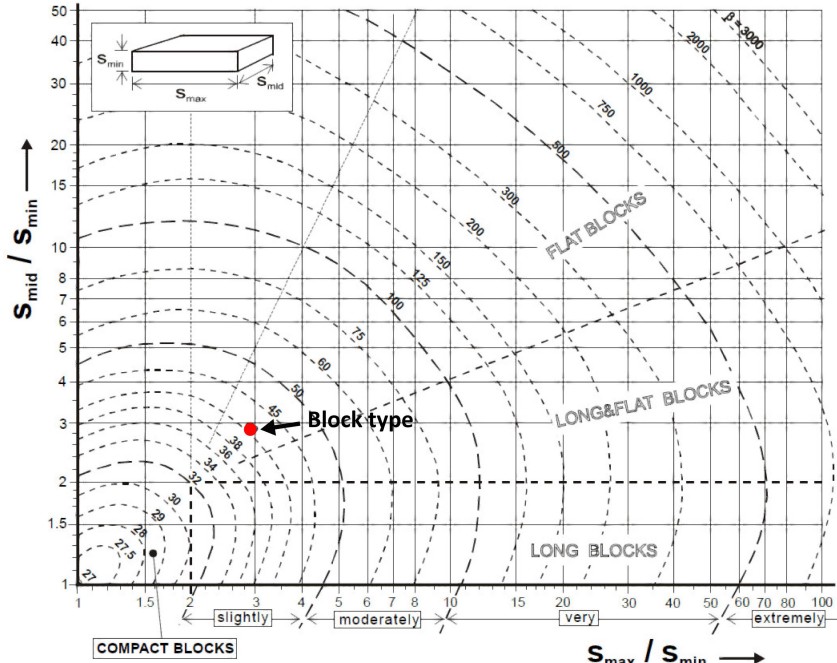

**Figure 13.** Step 4: semi-automatic estimation of block shape and volume of the experimental dataset (modified after 70).

### 3.2. Application to the Case Study

The results of the mean strike, spacing, trace lengths, and persistence, both for the MATLAB routine and field surveys, are shown in Figures 14 and 15. Three scanlines at different positions were traced on both the trace maps of JS1 and JS2 to investigate the effect of scanline positions on the estimation of the normal set spacing (Figure 16), as well as the fracture density and intensity maps (Figure 17). A sensitivity analysis was performed to

estimate the effects of the sample size and number of circles to calculate the persistence, trace length intensity, trace intensity, and trace density (Figure 18). An equivalent block volume, equal to 0.03 m$^3$, was obtained from the spacings of the two main discontinuity sets and the mean layer thickness (S0, calculated on exposed subvertical surfaces in the point cloud of the rock mass), using Equation (6). The reported volumetric joint count is 9.94 m$^{-1}$, while the block shape factor (β = 28.98) corresponds to a compact, slightly flat block shape, in agreement with field observations. The results of the Matlab routine are reported in Table 7.

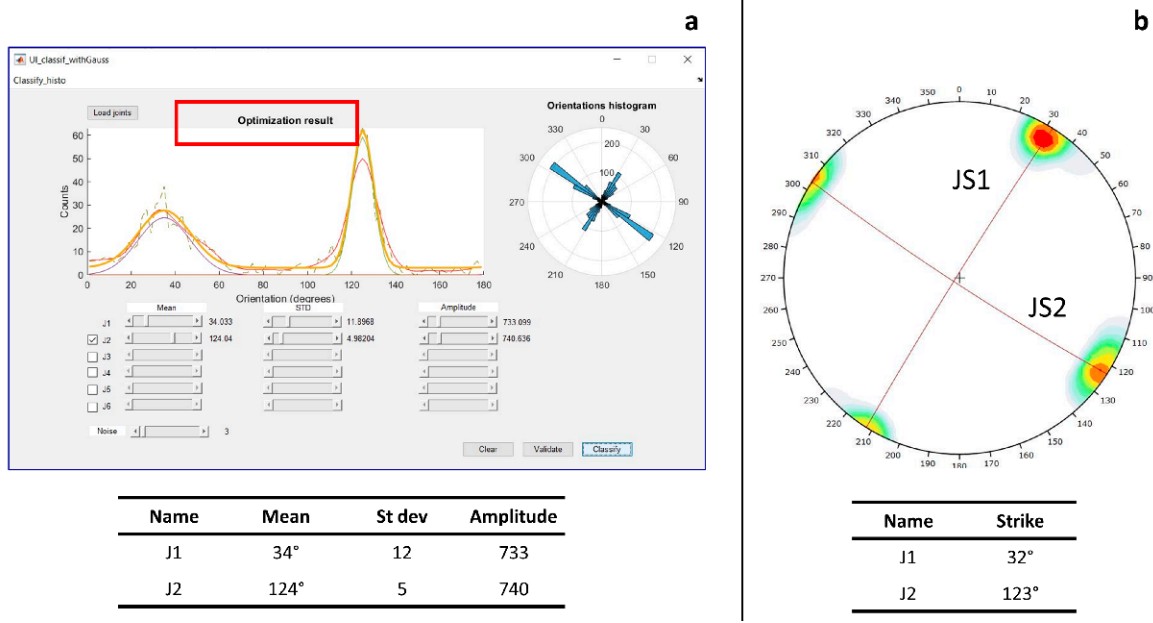

**a**

| Name | Mean | St dev | Amplitude |
|------|------|--------|-----------|
| J1 | 34° | 12 | 733 |
| J2 | 124° | 5 | 740 |

**b**

| Name | Strike |
|------|--------|
| J1 | 32° |
| J2 | 123° |

**Figure 14.** Results of the semi-automatic classification of the traces into joint sets (**a**) and of the data obtained from on-site geostructural and geomechanical surveys (**b**).

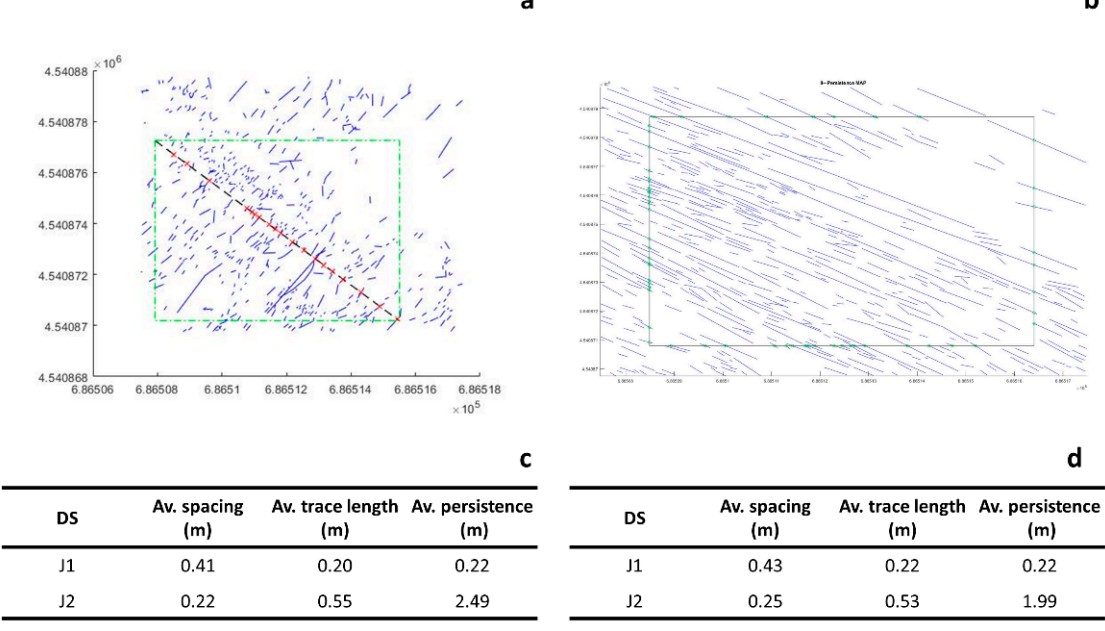

**a**

**b**

**c**

| DS | Av. spacing (m) | Av. trace length (m) | Av. persistence (m) |
|----|------|------|------|
| J1 | 0.41 | 0.20 | 0.22 |
| J2 | 0.22 | 0.55 | 2.49 |

**d**

| DS | Av. spacing (m) | Av. trace length (m) | Av. persistence (m) |
|----|------|------|------|
| J1 | 0.43 | 0.22 | 0.22 |
| J2 | 0.25 | 0.53 | 1.99 |

**Figure 15.** Results of the linear scanline and persistence methods. (**a**) Graphical representation of the scanline generated perpendicularly to J1; (**b**) graphical representation of the rectangular window generated for the calculation of the persistence of J2; (**c**) table of the parameters calculated with the MATLAB routine for J1 and J2; (**d**) table of the parameters measured in the field by means of conventional geostructural and geomechanical surveys.

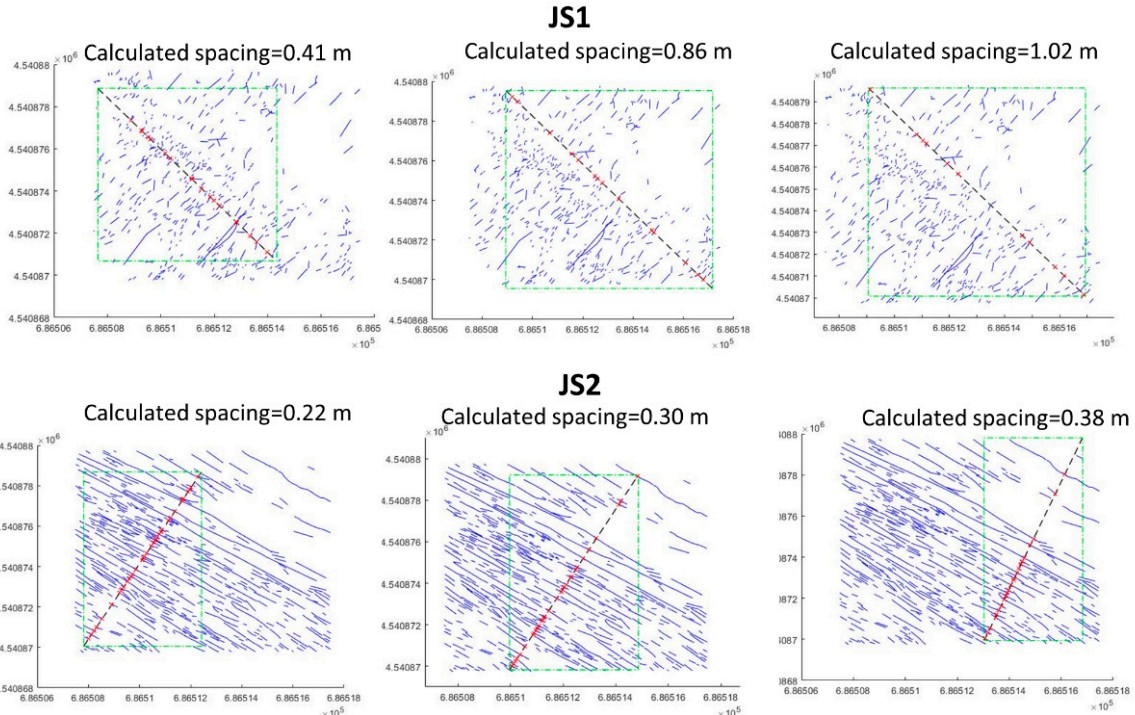

**Figure 16.** Normal set spacings calculated with different scanline locations.

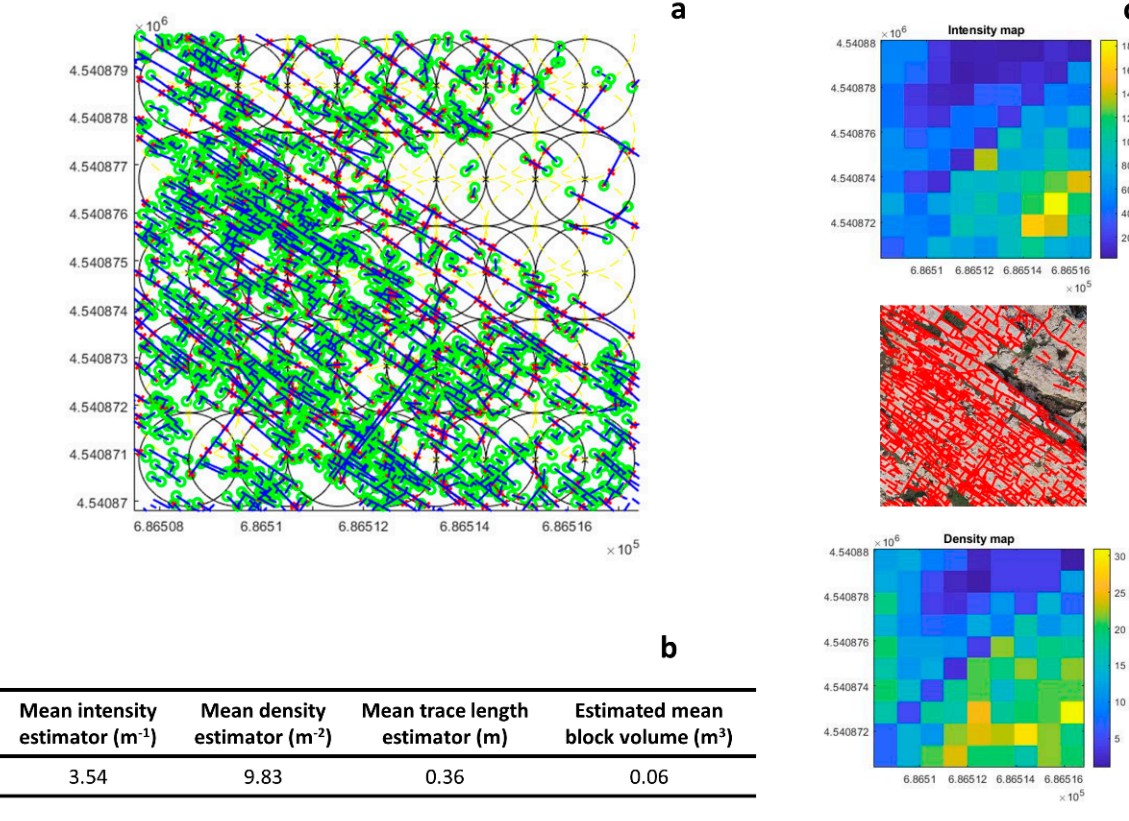

| Mean intensity estimator (m⁻¹) | Mean density estimator (m⁻²) | Mean trace length estimator (m) | Estimated mean block volume (m³) |
|---|---|---|---|
| 3.54 | 9.83 | 0.36 | 0.06 |

**Figure 17.** Evaluation of the jointing degree of the study area. (**a**) Graphical representation of the circular windows' method; (**b**) resuming table of the calculated mean intensity, density, and trace length estimators and block volume; (**c**) intensity and density maps of the analyzed area.

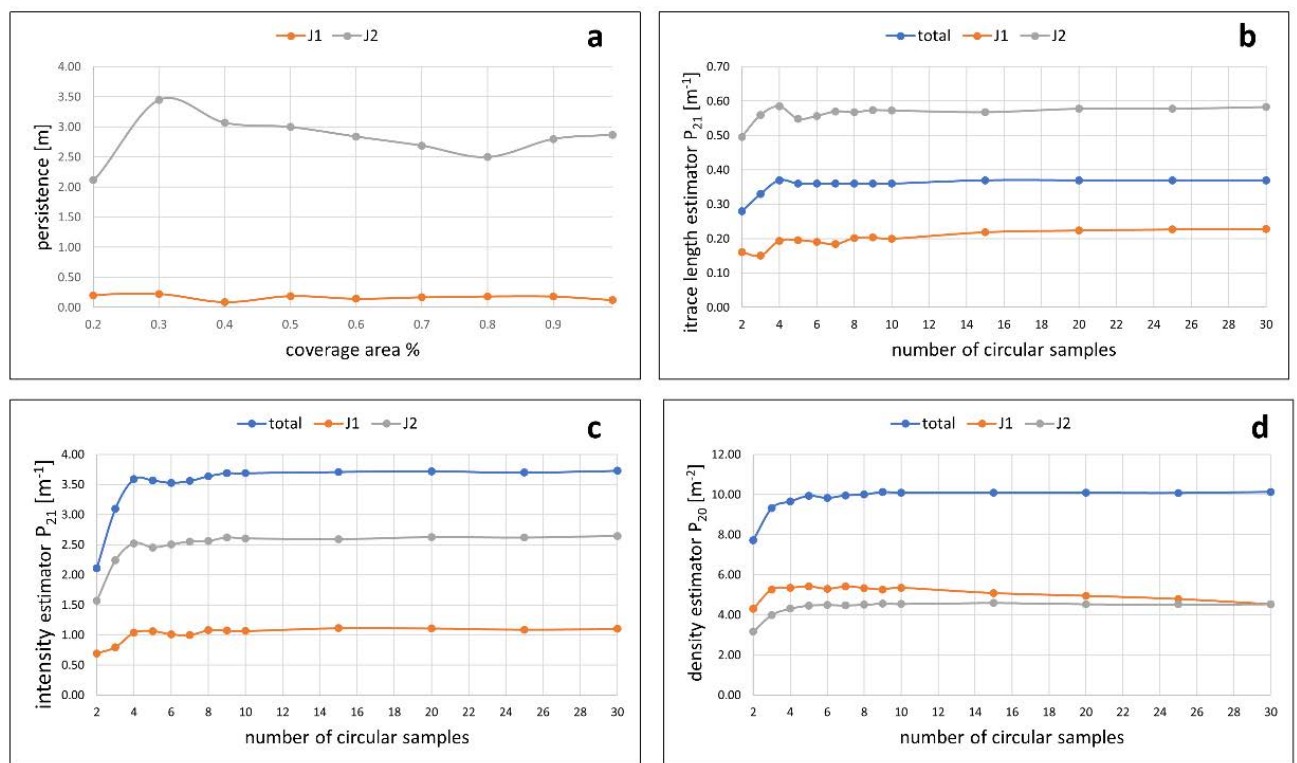

**Figure 18.** Sensitivity analysis to estimate the effects of the sample size and number of circles to calculate the persistence (**a**), trace length intensity (**b**), trace intensity (**c**), and trace density (**d**).

**Table 7.** Characterization of the discontinuity sets and of the jointed area with the MATLAB routine.

| **Characterization of Discontinuity Sets** | | | | | | |
|---|---|---|---|---|---|---|
| **Identified DS** | **Strike** | | | **Normal Spacing (m)** | **Mean Trace Length (m)** | **Mean Persistence (m)** |
| | **Mean Strike (°)** | **St. Dev. (°)** | **Amplitude** | | | |
| J1 | 34 | 12 | 733 | 0.41 | 0.20 | 0.22 |
| J2 | 124 | 5 | 740 | 0.24 | 0.55 | 2.49 |
| **Characterization of the Joint Network** | | | | | | |
| **Intensity Estimator $P_{21}$ (m$^{-1}$)** | **Density Estimator $P_{20}$ (m$^{-2}$)** | **Trace Length Estimator (m)** | **Volumetric Joint Count $J_V$ (m$^{-1}$)** | **Block Volume $V_B$ (m$^3$)** | **Block Shape Factor β** | **Block Shape** |
| 3 54 | 9.83 | 0.36 | 9.94 | 0.06 | 28.98 | compact |

Moreover, the routine was applied on the trace maps of sectors A, B, D, and E to identify potential differences in the number of discontinuity sets, as well as their mean strikes. Figure 19 shows the differences between the mean joint sets detected in sector C and sector B.

The statistics of strike, spacing, and trace length of the joint sets extracted from the MATLAB routine (Figure 20) were used to generate a realistic Discrete Fracture Network model of the Calcare di Bari Fm along the cliff located at Lama Monachile site. The data obtained from sector C were considered to be representative for the cliff after validation in the field (Figure 21).

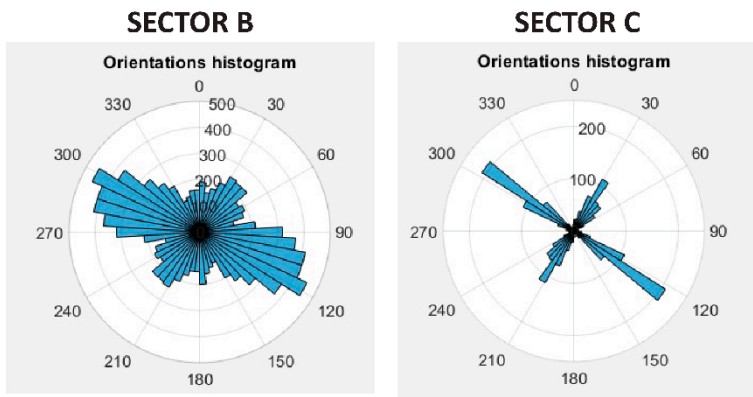

**Figure 19.** Rose plot of sector B (**left**) and sector C (**right**).

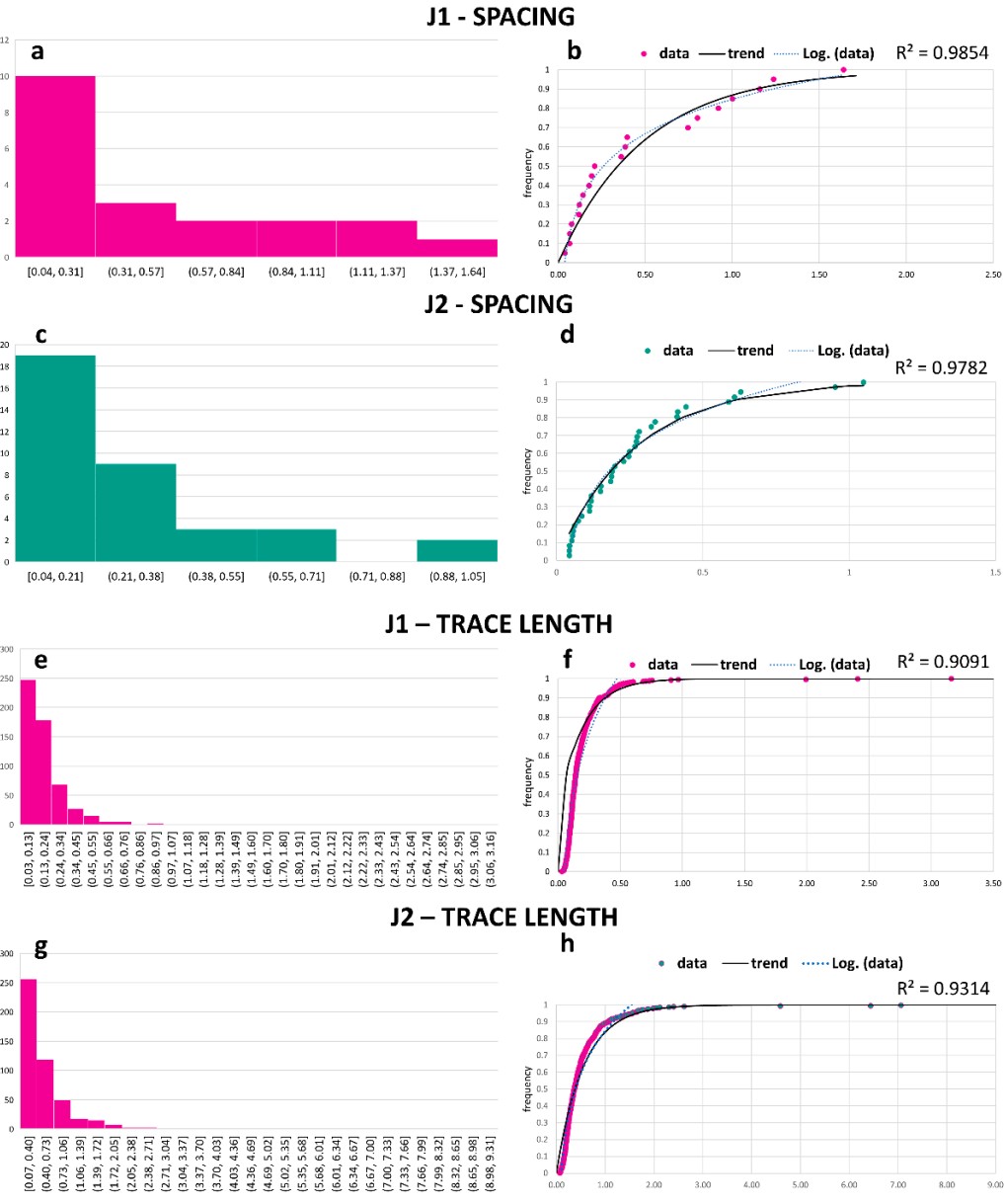

**Figure 20.** Histograms and cumulative frequencies of the spacing (**a–d**) and trace length (**e–h**) of the main joint sets for the generation of the Discrete Fracture Network Model.

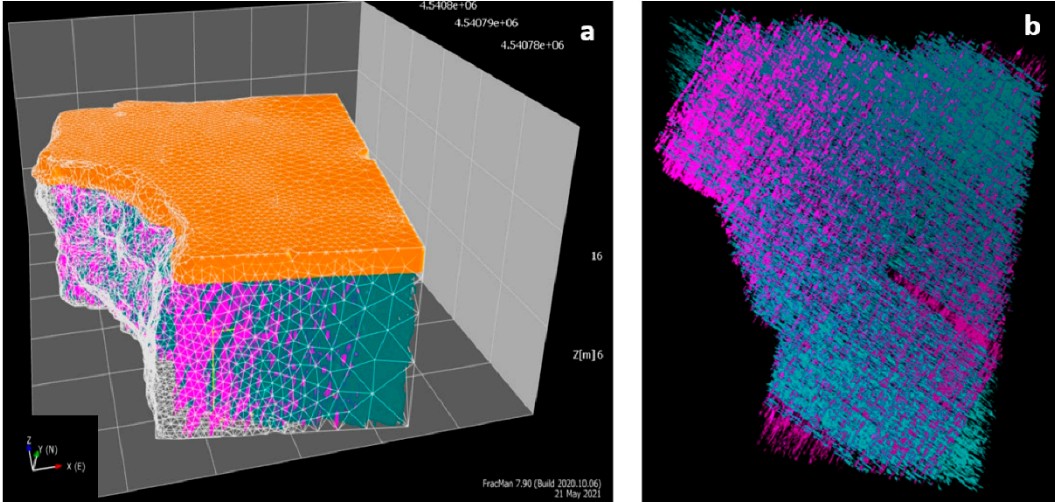

**Figure 21.** (**a**) DFN model created with the software FracMan for a sector of the rock cliff at Lama Monachile site, located at a distance of about 90 m from the study area. The fracture generation was limited to the Calcare di Bari Fm., which was the object of a specific geostructural survey. (**b**) Top view of the discontinuities generated in the DFN model of the Calcare di Bari Fm. at Lama Monachile site. The joint sets were generated using the results of the quantitative analysis of the discontinuities carried out at Pietra Piatta site. J1 and J2 are, respectively, colored in magenta and green.

## 4. Discussion

With regards to the synthetic dataset, the Gaussian fitting method recognized 3/3 discontinuity sets, with a difference of 2° for the strike of each discontinuity set and up to 1° for the standard deviation (Figure 5). On the contrary, some difficulties were found for the classification by means of the Hough transform method, because discontinuities with too similar orientations create unprecise clusters in the theta-r diagram (Figure 7). We found out that the histogram fitting method for classification is reliable for a high number of traces (at least 50), since it is based on statistic procedures, regardless of the orientations of the discontinuity sets. The Hough transform method is recommended for lower numbers of discontinuities, provided that the mean strikes of the sets are not too close (Figure 22).

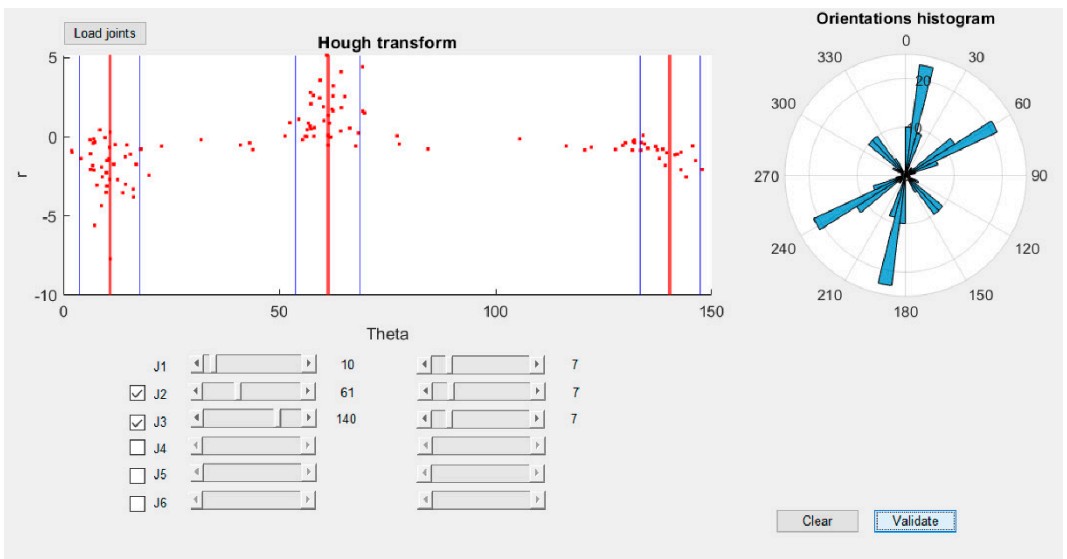

**Figure 22.** Correct classification by means of the Hough transform for three synthetic discontinuity sets with not too close orientations. The mean strike used for generating the synthetic dataset were 10°, 60°, and 140°.

Based on these observations, we processed the traces of the study site with the Gaussian fitting method and achieved good results with respect to the geostructural-geomechanical analysis carried out on the study site. The mean strikes of the discontinuity sets J1 and J2 obtained from the MATLAB routine were 34° and 124°. These results were in agreement with the data collected by means of field surveys (32° and 123° for J1 and J2, respectively). The same standard deviation value of the strike of J2 was found both for the MATLAB routine and for the on-site surveys, while a difference of about 4° was detected for J1. The standard deviations differed 4.2° for J1 and 0° for J2. This difference can be related to a major dispersion of the poles of the discontinuities, which was not detected in the field because of the lower number of sampled discontinuities (50).

With regards to the spacing estimations, the mean values were 6 cm (J1) and 2 cm (J2), higher than the data measured in the field. Several tests were carried out in order to obtain results in agreement with the field observations because the calculated mean spacing can vary significantly with the position and strike of the scanline (Figure 16). In this perspective, we remark that it is fundamental to use a proper dataset as input: unmapped traces in the orthophoto or objects/vegetation covering the joints could determine fewer intersections between the traces and the scanline, leading to an overestimation of the spacing. In addition, the possibility to apply linear samples in different locations of the dataset may help to understand the spatial variability of the calculated parameter, to identify potential changes of the stress field, or to detect sectors with different geomechanical behavior.

It is remarkable that both the spacing and trace lengths of J1 and J2 followed an exponential negative distribution, indicated by very good R-Squared values (in the range of 0.91–0.98, Figure 20), as found by [8,75,92,93] for a variety of rock masses.

A difference of 0.5 m was found in the estimation of the mean persistence of J2. It is believed that this difference is attributable to the different sampling methods for its estimation. Indeed, the persistence of discontinuities is one of the most difficult parameters to estimate in the field [65] and, during the conventional geostructural surveys, this property was roughly approximated from the trace lengths, according to [8]. However, this approach can lead to size and censoring bias [79,94], thus undersampling smaller discontinuities and censoring surfaces, which extend beyond the sampling area. Indeed, since the traces belonging to J1 were not truncated or censored because both the majority of their terminations fell within the area sampled in the field, no difference was found with respect to the results of the MATLAB routine. Based on these observations, it is believed that the results of the MATLAB routine are more reliable because the estimation of the persistence by means of [67]'s method takes into account the number of discontinuities transecting and contained in the sampling window, thus avoiding censoring bias. However, care should be taken when choosing the size of the coverage area. As pointed out by [8], this method is not suitable when all the discontinuities transect the window (the persistence would be infinite) and when no discontinuities are contained in the window (the persistence would be zero). For this reason, the sampling window should be chosen so that at least one trace is located inside it. In addition, we observed how the persistence varied with different dimensions of the sampling area (Figure 18a): Although two datasets are not enough to identify potential mathematical relations, it is evident that significantly different results were obtained by changing the area covered by the sampling window, especially for long traces (e.g., J2 curve in Figure 18a). As a matter of fact, [79] observed that persistence can vary up to more than 50% of the real value by changing the rectangular sampling window location and size.

The additional parameters calculated from the routine such as mean intensity, density, and trace length estimators give useful information for the identification of more fractured zones, represented by the yellow pixels in Figure 17c. Moreover, the fracture abundance of one discontinuity set constituted by non-parallel, subparallel, or non-persistent traces can be expressed by means of fracture intensity [69]. In fact, the estimation of fracture abundance for non-parallel traces by means of spacing is rather unclear because the distance is not unique. With regards to the length, intensity, and density estimators, sensitivity

analyses allowed us to observe that more accurate results can be achieved by choosing about 6–8 circles: A minor number would give unreliable results, while a larger number could cause unnecessary long computational times (Figure 10b,d).

Concerning the block volume, this method relies on the assumption that the blocks are determined by the totally persistent discontinuities; therefore, the block volume could be underestimated. More precise results can be obtained directly from measurements on 3D point clouds, with respect to 2D analyses.

The script for the semi-automatic identification of discontinuity sets can be used in different zones of an orthophoto to identify potential deformation zones or changes of the stress field. Figure 19 depicts a shear deformation zone in sector B detected from two additional joint sets (with mean strikes N–S and E–W) and higher standard deviations of the strike of J1 and J2 with respect to sector C, which are the result of non-linear shear structures.

The presented approach helps to calculate the geometrical parameters of the discontinuity sets affecting a rock mass in a less time-consuming, more precise, and safer manner compared to the conventional geostructural and geomechanical surveys. However, a complete characterization of rock masses also requires information on roughness, wall strength, aperture, filling, and seepage of the discontinuities that cannot be estimated from 2D analyses. The determination of wall strength, nature of filling, and seepage requires direct measurements at the site, while only centimetric apertures could be measured on high-resolution remote sensing products [29]. In addition, recent advances in the literature proposed methods to determine the roughness/undulation of discontinuities from point clouds [29,35,95,96].

In this perspective, the MATLAB routine can contribute to identify the main discontinuity sets from discontinuity traces in low-relief rock masses characterized by discontinuities perpendicular to the strata and combined with point clouds, to localize accessible and representative surfaces in order to measure the non-geometrical properties by means of conventional techniques. Finally, the combination of geometrical and non-geometrical properties of discontinuities and point clouds or meshes can be used to create discrete models for quantitative stability analyses in rock masses by means of discontinuum approaches. It is specified that interactions among individual fractures or discontinuity sets (fracture topology) need to be defined to characterize rock mass permeability, especially when dealing with fault damage zones and reservoir modelling [41,97–103]. In this perspective, the MATLAB routine developed could be improved through the characterization of fracture nodes and terminations [104,105] to define the network connectivity and its influence on the rock mass physics [106,107].

## 5. Conclusions

This study illustrated a new MATLAB tool for 2D semi-automatic analyses of discontinuities from high-resolution orthophotos obtained by means of remote sensing techniques, aimed at characterizing rock masses. Although similar tools were presented in the literature [29,40], they are not publicly available or do not provide a complete description of discontinuities for rock mass characterization. Therefore, the routine was compiled by adapting and updating the standard methods (i.e., scanline, rectangular, and circular window sampling) for a complete and detailed analysis of discontinuity traces, which is preferable to 3D point cloud analyses for low-relief rock masses or man-made excavations, in a user-friendly digital environment.

The code was initially built on a synthetic dataset and successively tested and validated on a case study by comparing the results of the conventional geostructural and geomechanical surveys carried out in the same area. The routine was developed in the form of consecutive steps, which can be singularly run depending on the objective of the analysis. In addition, the calculations do not require high-performance computers but can be run on standard laptops in a few seconds.

A new feature allows us to semi-automatically identify and classify the mean discontinuity sets in a Graphic User Interface, by fitting Gaussians curves on the strike histograms of the traces or by using the Hough transform, according to the number and approximate orientation of the discontinuity sets. In addition, the normal spacing of the discontinuity sets can be calculated both for persistent and non-persistent joints. The procedure for the classification and characterization of the discontinuity sets, as well as the estimation of the jointing degree of the analyzed area, can be easily repeated in different parts of an orthophoto to identify potential changes of the mean discontinuity sets, implicating modifications in the stress field, which can be further investigated on site for structural analyses.

Future developments may concern the improvement of the unit block volume calculation through direct measurements on the areas delimited by discontinuities rather than from spacing and orientation of the main discontinuity sets. Moreover, an automatic-semiautomatic method for drawing the discontinuities on orthophotos could exceptionally decrease the time required for data pre-processing.

Eventually, future research topics could deal with the conversion of the proposed Hough Transform method in the 3D space to detect discontinuity surfaces from point clouds or triangulated surfaces. The results could be compared with the methods available in the literature to validate the technique's reliability or identify potential discrepancies [108–110]. In addition, the Gaussian fitting method could be implemented to have 3D histograms of the orientations (x = dip, y = dip direction, z = frequency) and a composite Gaussian surface instead of a curve, to detect discontinuity planes from 3D models. Finally, discontinuity sets' spacings could be calculated both from the 3D Hough Transform Method and from the 3D Gaussian fitting method and compared with each other.

**Supplementary Materials:** The QDC-2D routine and the instruction file are available online in the GitHub public repository at: https://github.com/charlottewolff/-QDC-2D, accessed on 5 December 2021.

**Author Contributions:** L.L. coordinated the multidisciplinary assembling, reviewed the literature, participated in the remote sensing acquisition and processing, mapped the discontinuity traces, and conceptualized the described workflow. E.W. developed the main structure of the routine and C.W. optimized it with inputs from M.J. and M.-H.D. G.F.A. validated and supervised the research activity. M.P. reviewed the draft and provided the research funding. All authors have read and agreed to the published version of the manuscript.

**Funding:** This research received no external funding.

**Data Availability Statement:** The datasets generated and analyzed during the current study are available from the corresponding author on reasonable request.

**Acknowledgments:** We thank Golder company (www.golder.com, accessed on 5 December 2021) for providing an academic license of FracMan–Geotechnical Edition used to generate the Discrete Fracture Network Model illustrated in Section 3.2.

**Conflicts of Interest:** The authors declare no conflict of interest.

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
