# Peer review of "QDC-2D: A Semi-Automatic Tool for 2D Analysis of Discontinuities for Rock Mass Characterization"

_remotesensing, doi:10.3390/rs13245086_

Round 1

Reviewer 1 Report

This paper presents and compares two techniques for the characterization and recognition of 2D fractures into DFN models. While most of the literature is devoted to DFN synthesis and creation, here the focus is on the analysis and characterization.

Overall, I think the paper has some merits, as it introduces an analysis that in my knowledge is novel but currently errs on the side of formalism, leaving out various notations and formalisms.

For instance, I see the use of the abbreviation DFN far from their complete name (Discontinuity Fracture Network), as well as the Hough transform is simply referred to as Duda's work, ignoring the evolution of tools and methods related to Hough definition (I doubt the algorithm used in this work is exactly Duda's one). Similarly, also the definition of persistence is omitted. I agree that this contest is probably unique but there are other definitions in the literature and therefore I think it is opportune to recall the definition used in this work.

Moreover, I'd suggest referring to some state-of-the-art paper, for instance:

Qinghua Lei, John-Paul Latham, Chin-Fu Tsang, The use of discrete fracture networks for modelling coupled geomechanical and hydrological behaviour of fractured rocks, Computers and Geotechnics, Volume 85, 2017

A. Lisjak, G. Grasselli, A review of discrete modeling techniques for fracturing processes in discontinuous rock masses, Journal of Rock Mechanics and Geotechnical Engineering, Volume 6, Issue 4, 2014,

To complete the analysis, I think the authors should investigate also the role of fracture topology in rock characterization and behaviour.

Regarding the results shown in the paper, I'd expect to see more measures related to the pattern recognition, such as the Dice index and also, for the measures reported in the paper, the explicit indication of the "ideal" value of reference.

Finally, in the conclusions and future works, I would appreciate some hints and discussions on how these techniques can be extended to 3D fractures., for instance considering the extension of the Hough to plane recognition in the space.

Author Response

  • Comment 1: This paper presents and compares two techniques for the characterization and recognition of 2D fractures into DFN models. While most of the literature is devoted to DFN synthesis and creation, here the focus is on the analysis and characterization.

Response: Thank you for the positive feedback. We remark that our Matlab routine was developed to characterize discontinuities from orthophotos and that the results can be used as input for other softwares to generate DFN models to study, for instance, the geomechanical and hydrological behaviour of rock slopes. With regards to the two mentioned techniques, we highlight that they were proposed to classify discontinuities from trace maps into discontinuity sets.

  • Comment 2: Overall, I think the paper has some merits, as it introduces ananalysis that in my knowledge is novel but currently errs on theside of formalism, leaving out various notations and formalisms.

For instance, I see the use of the abbreviation DFN far from their complete name (Discontinuity Fracture Network), as well as the Hough transform is simply referred to as Duda's work, ignoring the evolution of tools and methods related to Hough definition (I doubtthe algorithm used in this work is exactly Duda's one). Similarly, also the definition of persistence is omitted. I agree that this contest is probably unique but there are other definitions in the literature and therefore I think it is opportune to recall the definition used in this work.

 Response: We thank you for aknowledging the uniqueness of our work and for identifying these inaccuracies. We provided to add references to the Hough transform method and to specify the literature definitions used for this study. The complete name of DFN is Discrete Fracture Network, as also mentioned in the suggested papers.

  • Comment 3: Moreover, I'd suggest referring to some state-of-the-art paper.

Response: We cited the suggested papers, as well as other relevant state-of-the art works.

  • Comment 4: To complete the analysis, I think the authors should investigate also the role of fracture topology in rock characterization and behaviour.

Response: Thank you for this suggestion. It would have been interesting to explore this aspect. However, this topic is outside the scope of our paper and would require significant changes within the code, at incompatible times with those required for the submission of the reviewed manuscript (also for previous commitments of some of the Authors, at present abroad for several weeks). We agree with you that fracture topology is important to fully characterize fracture networks and could be an interesting subject for future studies, as reported in the discussions section.

  • Comment 5: Regarding the results shown in the paper, I'd expect to see more measures related to the pattern recognition, such as the Dice index and also, for the measures reported in the paper, the explicit indication of the "ideal" value of reference.

Response: we are sorry to comment that we do not understand how the Dice index could be applied in this case, because we did not apply image analysis in our work. Moreover, as we used different techniques for fracture characterization (field measurements and Matlab routine), there are not reference values, but probability distributions, as reported in the text and images. If you believe that these implementations are essential for our work, we ask you to kindly indicate some examples.

  • Comment 6: Finally, in the conclusions and future works, I would appreciate some hints and discussions on how these techniques can be extended to 3D fractures., for instance considering the extension of the Hough to plane recognition in the space.

Response: We discussed the proposed topics in the conclusion section.

Reviewer 2 Report

Dear authors,

There are many minor mistakes, please revise this:

Line 13: delete "of"

Line 14: Delete To

Line 14: Delete The before "data accuracy"

Line 17: providing, not "provide". It appears that your sentence or clause uses an incorrect form of the verb "provide"

Line 20: delete this phrase word "to our knowledge" , put it right after "Nevertheless"

Line 23: delete "Also" it is unnecessary in this sentence.

Line 25: delete "from which a". Need to separate this sentence into two sentences. 

Line 33: it should be: procedure's reliability "

Line 41: delete "for the characterization of" , instead by " to characterize"

Line 44; Please add more an article "a" before "reasonable" 

Line 63: delete "by means of" change to "utilizing" 

Line 77: delete "allows to identify" change to "identifies"

Line 106: delete "The coast is formed by platforms and cliffs" , change to this sentence: Platforms and cliffs form the coast up to 20 m high, linked by embayments constituted by coastal erosion deposits (pocket beaches)

Line 113; Delete "constituted by" change to "composed of". Need to add more "which are" before "discontinuously"

Line 122: Please delete " In coastal karst settings, these ", rewrite this sentence, you can refer this one:
These are among the most frequent geological hazards in coastal karst settings and are partly favored by the diffuse presence of karst conduits and caves, further weakening the carbonate rock mass [46–49]. 

Line 165: rewrite this sentence, delete " software", delete :model of the rock mass":
The SfM technique was carried out with the Agisoft Metashape Professional [50] to process the images and obtain a 3D rock mass model.

Line 200: delete "aimed at collecting" change to "to collect".

All the equations need to mention in the relevant content. 

Line 553: These two sentences are hard to follow, please rewrite this.

Line 563: delete "as input a proper dataset" , change to "a proper dataset as input.

The other parts are good written. 

Author Response

Response: Thanks for the time spent to check our text. We made all the suggested modifications and checked that all the equations mention in the relevant content.

Reviewer 3 Report

  1. Line 31~33, the sentence is too long to read;
  2. Line 79, there should be citations.
  3. Hyperspectral LiDAR is feasible to this application, please kindly cite the papers:  [1] Shao, H., Chen, Y., Yang, Z., Jiang, C., Li, W., Wu, H., ... & Hyyppä, J. (2019). A 91-Channel Hyperspectral LiDAR for Coal/Rock Classification. IEEE Geoscience and Remote Sensing Letters17(6), 1052-1056. [2] Shao, H., Chen, Y., Yang, Z., Jiang, C., Li, W., Wu, H., ... & Hyyppä, J. (2019). A 91-Channel Hyperspectral LiDAR for Coal/Rock Classification. IEEE Geoscience and Remote Sensing Letters17(6), 1052-1056.

Author Response

  • Comment 1: Line 31~33, the sentence is too long to read

Response: we fixed the reported sentence.

  • Comment 2: Line 79, there should be citations.

Response: we added relevant references.

  • Comment 3: Hyperspectral LiDAR is feasible to this application, please kindly cite the papers: [1] Shao, H., Chen, Y., Yang, Z., Jiang,C., Li, W., Wu, H., ... & Hyyppä, J. (2019). A 91-ChannelHyperspectral LiDAR for Coal/Rock Classification.

Response: thanks for suggesting this paper, we provided the suggested citation in Section 1.

Round 2

Reviewer 1 Report

Overall, I am satisfied with the review work provided by the authors. I think that have satisfactorily replied to my requests and added further references and discussions.

I apologize for the confusion created with the request to evaluate the Dice index. Actually, I had in mind it as a coefficient (and it is also known as Dice-Soeretsen). It can be evaluated on any type of data, once the true positive (TP), false positive (FP), and false negative (FN) scores are defined. If classification was possible I assumed these values were known. In that case, the Dice-Soerentsen coefficient (DSC) can be written as

.